# Factors Influencing Viewing Behavior in Live Streaming: An Interview-Based Survey of Music Fans

**Minjeong Ham and Sang Woo Lee ***

Graduate School of Information, Yonsei University, 50 Yonsei-ro, Seodaemun-gu, Seoul 03722, Korea; mnjnghm@gmail.com

* Correspondence: leesw726@yonsei.ac.kr

**Abstract:** V Live is a live-streaming service made by South Korean IT company in August 2015. The service provides diverse video contents specific to entertainment content. Most of V Live users are K-pop fans, and they actively express emotions on V Live content by writing comments, pressing "hearts", and sharing video content. Based on Uses and Gratifications theory, this study investigated why people use live streaming service, and the factors influencing users' viewing behavior in live streaming. We conducted an in-depth interview with V Live users. Based on the results of the interview, an online survey was conducted. As a result, six factors—"Interpersonal relationship motivation", "Social presence motivation", "Celebrity support motivation", "Celebrity presence motivation", "Social interaction motivation", and "Differentiation motivation"—were derived as motivations to use V Live. While "Social presence motivation" and "Differentiation motivation" among V Live use motivations that have been shown to mediate the relationship between fans' fanship and V Live viewing time, all motivations using V Live have been shown to mediate the relationship between fans' fanship and V Live viewing participation.

**Keywords:** fanship; V Live; live streaming service; participation

---

## 1. Introduction

The K-pop boy band BTS held a concert at Wembley Stadium in London, the UK, in June 2019, which was broadcast live on Naver's live streaming service (LS), V Live. Even though V Live's live streaming of the BTS concert was paid content that cost users an additional USD 27, Naver earned approximately USD users from 3,750,000 to 140,000 [1]. V Live exceeded 85 million cumulative downloads in the fourth quarter of 2019, and its paid channel was evaluated as a successful new business model [2].

V Live, which started in August 2015, has actively provided live streaming content and self-produced content related to K-pop stars and targeted fandom. To lower the language barrier for the convenience of worldwide fans, V Live supports "V Fansubs", a real-time subtitle service that enables a transnational network of K-pop stars and K-pop fandoms [3].

LSs like V Live are real-time broadcasting platforms provided in a wireless network environment rather than a traditional broadcasting network. There are three characteristics of LSs: first, real-time interactive communication between a streamer (a person who live-streams) and a user is possible; second, there are no restrictions on time and space; and third, anyone can be a streamer [4]. Leading global LSs include Twitter Periscope and Amazon Twitch from the US, and Douyou TV from China; popular South Korean services include Afreeca TV and V Live. However, unlike other services, V Live is not open to "everyone" to produce video content. Through contracts with entertainment management agencies, V Live exclusively grants K-pop stars permission to start a channel.

Real-time interactive communication on the LS creates continuous interaction between streamers and users, building a closer relationship [5]. For example, the user "consumes", a passive act of watching video content or reading other user's comments, or "participates", in an active act of directly commenting or sharing and making video content [6]. In particular, the act of pressing "like" by users reflects their positive emotions toward the video content [7], and the virtual money contributed by users to streamers is a means of expressing the user's preference for the video content being watched, or that the user is a fan of the streamer [8]. By participating, users support streamers or prove their popularity. While the "like" option on Facebook or YouTube can be clicked only once, the "heart" on V Live can be pressed indefinitely. The number of "hearts" on V Live is used as a measure of a celebrity's popularity and also induces ranking and competition among fans [3,9]. A V Live user taps the "heart", which can be pressed indefinitely, actively, and continuously—before, during, and after watching the video content. This has led to a new fandom culture known as "heart labor [9]".

Studies on LSs that have enabled real-time interactive communication between streamers and users across time and space are actively underway. In particular, user-level research is diverse. Hilvert-Bruce et al. (2018) examined the motivation of Twitch users and their participation while viewing [10]. Chen & Lin (2018) investigated social network service (SNS) users' attitude towards and continuance usage on the LS [11]. To the best of our knowledge, there are no studies on V Live users' motivation, usage, and participation. Studies on V Live have focused on the video content provided by the platform. Jing et al. (2018) investigated the relationship between the type of music content and number of viewers on V Live [12]. Ham & Lee (2020) examined the factors influencing the popularity of video content on V Live [13].

This study investigates the motivation that drives V Live use, if there is a difference in the viewing time and extent of participation while viewing V Live depending on users' motivation, and the reasons for the same. V Live users' motivation will be different from users of other LSs. On V Live, the streamers are K-pop stars, and video content is produced in collaboration with Naver and other entertainment management agencies who invest capital and plan such content. In addition, this study analyzes the previously unexplored relationship between the motivation of V Live users and their participation while viewing to broaden the understanding of K-pop fandoms, who are the key consumers in the entertainment industry.

## 2. Literature Review

### 2.1. Motivation for V Live: Uses and Gratification Theory (UGT)

According to the UGT, users actively use media to have their specific needs met (Katz et al. [14]). The UGT offers a framework for understanding the choices and participation of media users.

Kim et al. (2016) explain that Afreeca TV users used the platform to seek information, interact with broadcast jockeys (BJs) (similar to streamers), watch video content that is differentiated from existing broadcast content, and to habitually spend time [15]. In particular, as BJs are streamers of Afreeca TV, the motivation to directly interact with a BJ is noteworthy as an exclusive feature of Afreeca TV.

Sjöblom & Hamari (2017) found that Twitch, a game-specialized LS, is used to seek game-related information such as game strategies or new games, enjoy game-related content, and build an emotional relationship with Twitch streamers or other users [4].

Hilvert-Bruce et al. (2018) focused on social motivations of Twitch users and found the platform was used to meet new people, participate in social interactions, form social bonds, gain unity as members of the community, and resolve interpersonal offline difficulties [10].

Hou et al. (2020) investigated that YY, the Chinese live-streaming service, is used to directly interact with streamers and other users, display social status, have fun and watch the appearance of streamers [16].

In summary, users of LSs have motivations to seek information or fun, to form ties with streamers or other users, and to watch previously unseen video content. Whenever new media such as TV, Internet,

and LSs have appeared, studies have largely focused on the media user's motivation to use the new media based on the UGT. Although it has been almost six years since V Live commenced services and its size and profits have steadily risen, extant research mainly deals with LSs such as Periscope, Twitch, and Afreeca TV. There is a lack of user-level research on V Live. While Hilvert-Bruce et al. (2018) focus on social motivations, this study aims to identify overall motivations to use V Live [10]. In other words, this study will investigate the motivations of V Live users to build social relationships with other users, use differentiated services, and support or communicate with K-pop stars.

By referring to prior studies that examined motivation for using LSs, we organized the survey items, and in particular, conducted in-depth interviews with V Live users to investigate their motivation for specifically using V Live. Thus, the research question to empirically examine the motivation of V Live users is as follows:

RQ1. What is the Motivation for Using V Live?

### 2.2. V Live User: Fan

A fan is an individual who is enthusiastic about celebrities. Fandom refers to a mass of fans, and fanship is the psychological state of a person who is passionate about something [17,18].

Fanship does not simply remain in a psychological state—it also induces psychological attachment and active participation [19,20]. For example, the higher the fanship for an influencer on an SNS, the greater the intention to maintain an online relationship with the influencer and to purchase products sold by them [19]. Higher fanship for a university sports team (i.e., the more passionately the fans like the sports team) corresponds to more sports-related SNS use for various purposes like seeking sports-related information and pleasure. Furthermore, the higher the motivation, the higher the satisfaction with university life [21]. In other words, fans who have fanship for a certain entity use social media for various reasons, such as obtaining information on their favorite target or communicating with other fans. Also, users who have a higher affective commitment to the streamer engaged more in live-streaming content [22] and intend more to donate to the streamer [23]. The results of social media use on the intention to sustain relationships, purchase products, or the extent of satisfaction differ depending on how enthusiastically fans love the target (the extent of fanship for the target). Users of LSs are video content viewers and fans of the streamers [24]. As V Live focuses on K-pop stars' video content, V Live users are fans of K-pop and K-pop stars [3,12,13]. Studies indicate that fans with fanship for a certain target may use V Live for various reasons such as obtaining K-pop star-related information or communicating with K-pop stars or other fans, and the motivation for using V Live may vary depending on the extent of fanship. Moreover, when using V Live, users can communicate directly with K-pop stars and other fans, and as a result, feel satisfied as members of the fandom.

### 2.3. Viewing and Participation on V Live

LS users consume video content in two ways: first, by simply watching the video content, and second, by participating in video content by clicking "hearts" or commenting while watching. Khan (2017) defined the viewing behaviors of YouTube users, wherein "consumption" refers to simply watching videos and reading other user's comments, and "participation" includes clicking, "likes", writing comments, and sharing or posting video content [6]. In particular, when a user clicks "like", it reflects their positive emotions about the video content [7]. The virtual money contributed by the user to a streamer is a means of expressing the user's preference for the video content being watched, or that the user is a big fan of the streamer [8]. Commenting on video content and clicking "hearts" are expressions of satisfaction on the video content by V Live users; the latter is similar to using Facebook's "like" button. V Live users simply watch live streaming content or also click the "heart", comment, or share video content while watching. In other words, positive emotions on V Live content or K-pop stars are expressed by viewing time and the extent of participation.

As mentioned before, Facebook's "like" can be pressed only once, while V Live's "heart" can be pressed indefinitely. K-pop stars' fans, the main users of V Live, are known to be the most active users

in terms of clicking the "heart" [3], and "hearts" are used as a measure of the celebrity's popularity, while also inducing ranking and competition among fans [3,9].

Unlike other LSs, V Live ranks its top 10 global artists and provides rewards. "Global Artist Top 10" is an event in which only the top 10 teams out of almost 1,500 K-pop stars' channels are selected. The top 10 teams are determined by the number of views, "hearts", comments, and fans' attendance score. The desire to present an award to K-pop stars or see K-pop celebrities at awards ceremonies influences fans to watch V Live content or participate in V Live content by clicking the "heart", commenting, or sharing content. Moreover, the desire to contribute to the content of a favorite K-pop star can encourage fans' participation while watching.

V Live users suggest certain actions or opinions to K-pop stars through comments, and K-pop stars create video content in real-time based on these comments, which can be described as K-pop stars and users collaborating to create video content for V Live.

Furthermore, the viewing time and participation in LS content are affected by the motivation to use LS. Hilvert-Bruce et al. (2018) found that Twitch users who were more motivated by having fun (Entertainment motivation), seeking information, and social interaction tend to use Twitch more frequently [10]. Furthermore, Twitch users who have higher levels of motivation for social interaction and a sense of community subscribe to the Twitch channel for longer periods and pay higher sums of money to the streamers. Hu & Chaudhry (2020) suggested that three bonds such as financial bonds, social bonds, structural bonds are the major factors inducing consumer's engagement in e-commerce live-streaming service. Financial bond refers to the economic benefits that consumers gain from live-streaming, social bond refers to the real-time direct interaction between streamer and consumers, and structural bond refers to the utility gained from the convenience of the platform such as ease of use [22].

### 2.4. Mediating Effect of V Live Use Motivation between Fanship, Viewing Time and Participation

Viewing time and participation in video content can vary depending on the motivation for using LS, and motivation can be influenced by the extent of fanship. For example, students with higher motivation for entertainment on sports-related SNSs tended to deliver information on sports-related SNS, and students with higher motivation to check their social position ultimately increased satisfaction with college life. Likewise, the motivation to use V Live will vary depending on users' fanship, and the motivation for using V Live may also impact viewing and engaging with the platform.

This study presents the relationship between V Live user's fanship, motivation for use, and V Live viewing time and participation during viewing. The extent of fanship of the V Live user is set as an independent variable, motivations for use are set as the mediator, and viewing time and participation during watching are set as dependent variables. The research questions and research models are posed as follows:

RQ2. Does motivation for using V Live mediate the relationship between the extent of V Live user's fanship and viewing time?

RQ3. Does motivation for using V Live mediate the relationship between the extent of V Live user's fanship and the extent of participation during viewing?

## 3. Methodology

### 3.1. In-Depth Interview

An in-depth interview was conducted with eight V Live users on 7 May 2019 to identify the motivation for use specific to V Live. The interview was targeted at V Live users who met three conditions: (1) use of V Live for more than two years, (2) subscription to channels of his/her favorite K-pop stars, and (3) experience of offline fan activities (e.g., concert) and online fan activities (e.g., music streams).

Eight interviewees were asked questions about their age, the period of using V Live, the K-pop star's channel that they had subscribed to, and the period of their activities as fans. Besides, they were encouraged to describe differences between V Live and other video services. As reported in Table 1, the average age of the interviewees was 32 years, the average period of fan activities was approximately 6.25 years, and the average period of using V Live was 36.25 months (approximately three years). Most interviewees were fans of K-pop stars like BTS, Kang Daniel of Wanna One, and Nu'est.

**Table 1.** Profiles of In-depth Interviewees.

| Code | Age | Period of V Live Use | Subscribing Channel of K-Pop Star | Period of Fan Activity |
|---|---|---|---|---|
| A | 27 | 3 years and 6 months | BTS | 6 years |
| B | 36 | 3 years and 6 months | Wanna One | 2 years |
| C | 35 | 3 years and 6 months | BTS, KIM JUNSU | 2 years, 10 years |
| D | 36 | 3 years and 6 months | KIM JUNSU | 10 years |
| E | 32 | 3 years | Wanna One | 2 years |
| F | 26 | 2 years | Nu'est | 2 years |
| G | 34 | 2 years and 8 months | BTS, KIM JUNSU | 10 years, 4 years |
| H | 32 | 2 years and 6 months | Wanna One | 2 years |
| Average | 32 | 36.25 months | | 6.25 years |

Two main reasons to use V Live emerged through the in-depth interviews. First, V Live users felt as if they were in the same place with their favorite K-pop star while watching the content. In existing video content, K-pop stars are seen in make-up and are well-dressed, while on V Live, they communicate with fans more naturally from their homes or cars, and fans feel as if they are face-to-face and in the same place as the celebrities they are watching. This study labels this existing motivation factor for use as "Social presence", and the motivation for using V Live to watch K-pop stars in their natural looks and feel their presence as "Celebrity presence".

The second reason is that by clicking "hearts", commenting or sharing video content, V Live users can contribute toward pleasing their favorite K-pop stars and receive awards such as the "Global Artist Top 10" when participation is higher. In other words, the motivation for using V Live is to prove the popularity of the K-pop star of their preference. This motivation to use V Live to boost the popularity of K-pop stars is labeled "Celebrity support".

*3.2. Survey*

In addition to the in-depth interview, an online survey of V Live users was conducted to examine the relationship between V Live users' fanship, motivation for using V Live, V Live usage, and participation in viewing. The survey was conducted for five days, from 17 May 2019 to 21 May 2019, by online survey firm Macromill Embrain. The questions for sample selection were, "How many times have you watched Naver V Live in the last month?" and "Do you use Naver V Live's star channel such as BTS, EXO?" only those who had used star channels at least twice a month were surveyed.

In all, 211 people responded to the survey, of whom 33.6% were male; 6.2% were in their teens, 38.4% were in their 20s, 36.5% were in their 30s, 16.1% were in their 40s, and 2.8% were in their 50s, and the average age of respondents was 31.87 (S.D. = 8.54).

For the period of V Live use, 19.0% responded "less than 6 months", 21.8% responded "more than 6 months to less than a year", 17.5% responded "more than 1 year to less than 1 year and 6 months", 12.3% responded "more than 1 year and 6 months to less than 2 years", 11.4% responded "more than 2 years to less than 2 years and 6 months", 2.4% responded "more than 2 and 6 months to less than 3 years", 8.1% responded "more than 3 years to 3 years and 6 months", and 7.6% responded "more than 3 years and 6 months".

The average weekly frequency of V Live access was 3.90 times (S.D. = 3.92). Regarding devices for V Live access, 83.4% used mobile devices such as tablet PCs or smartphones, and 15% used a PC.

*3.3. Measurement*

3.3.1. Fanship

Referring to Reysen & Branscombe (2010), the measurement items for fanship were: "I feel good when my favorite K-pop star is popular", "I will devote all my time to my favorite K-pop star if I can", "I spend a lot of money on my favorite K-pop star", and "I will feel hopeless if I can no longer like my favorite K-pop star" (reverse coding) [18]. All items were measured on a 5-point Likert scale.

3.3.2. Motivation for Using V Live

The items for "Interpersonal relationship (motivation to build social relations with other fans", "Social presence (motivation to feel the presence of other fans)", "Social interaction (motivation to directly interact with a K-pop star", and "Differentiation (motivation to use a differentiated video service, i.e., V Live)" were properly reconstructed referring to the prior studies [11,15,25]. The items for "Celebrity support (motivation to boost the popularity of a K-pop star)", and "Celebrity presence (motivation to feel the presence of a K-pop star)" were constructed on the basis of results of in-depth interviews. All times were measured on a 5-point Likert scale.

3.3.3. Viewing Time on V Live

The viewing time on V Live was calculated by multiplying the average weekly frequency of V Live access to viewing time of one V Live access. The average weekly frequency of V Live access was measured by the question, "How many times do you log on to Naver V Live on average in a week?" The respondents answered directly.

The viewing time of one V Live access was measured by the question, "For how long on an average do you watch live streaming content once you log on to V Live?" Respondents chose one from "Not at all", "Less than 15 min", "More than 15 min but less than 30 min", "More than 30 min but less than 45 min", "More than 45 min but less than 1 h", "More than 1 h but less than 1 and a half hours", "More than 1 and a half hours but less than 2 h", "More than 2 h but less than 2 and a half hours", "More than 2 and a half hours but less than 3 h", and "More than 3 h".

3.3.4. Participation in Viewing V Live

Participation in viewing V Live is the extent to which V Live users click "hearts", post comments, and share content with friends while viewing videos. Referring to Khan (2017) and Hilvert-Bruce et al. (2018), the participation was measured by the question "How much do you participate in viewing live streaming content on Naver V Live? [6,10]" The respondents answered each item: "When I watch live streaming content on V Live, I tend to comment", "I tend to click hearts", and "I tend to share it with friends". All items were measured on a 5-point Likert scale.

## 4. Results

*4.1. RQ1*

Research question 1 was used to investigate the motivation for using V Live, and an exploratory factor analysis was conducted using the Varimax method to verify the validity of the measurement items (see Table 2). Principal component analysis (PCA) was performed to extract the main components by reducing variables that were highly correlated to one single dimension. A total of six factors were extracted, including (1) interpersonal relationships, (2) social presence, (3) celebrity support, (4) celebrity presence, (5) social interaction, and (6) differentiation, as motivations for using V Live. In other words, V Live users use the service with the desire to build an interpersonal relationship with people or other fans (Interpersonal relationship), to feel as if the user and other fans are in the same place (Social presence), to support their favorite K-pop star (Celebrity support), to feel as if the user and the K-pop star are in the same place (Celebrity presence), to directly communicate with

their favorite K-pop star (Social interaction), and to watch video content that cannot be seen on other LSs (Differentiation).

**Table 2.** Results of Factor Analysis.

| | Inter-Personal Relation-Ship | Social Presence | Celebrity Support | Celebrity Presence | Social Interaction | Differen-Tiation |
|---|---|---|---|---|---|---|
| I get the opportunity to talk with the people/fans that are around. | 0.828 | | | | | |
| I can develop a relationship with the people/fans that are around. | 0.779 | | | | | |
| I can avoid being sidelined in conversations with the people/fans that are around. | 0.738 | | | | | |
| It feels like the other viewers who are also watching are in the same place as me. | | 0.772 | | | | |
| I feel like I am meeting with other viewers in person and talking to them. | | 0.852 | | | | |
| Other viewers who are watching together feel like they are next to me or in front of me. | | 0.793 | | | | |
| The video contents of my favorite K-pop star must achieve the record. (e.g., receiving over 100 million "hearts") | | | 0.837 | | | |
| To see my favorite K-pop star react when his or her content has achieved a certain record, such as 100 million hearts. | | | 0.820 | | | |
| I can prove my favorite K-pop star's popularity to other K-pop stars' fans. | | | 0.765 | | | |
| I can see the natural appearance of my favorite K-pop star. | | | | 0.842 | | |
| I can see my favorite K-pop star's latest image. | | | | 0.809 | | |
| I can ask my favorite K-pop star questions in real-time while live streaming. | | | | | 0.719 | |
| I can communicate with my favorite K-pop star. | | | | | 0.684 | |
| My favorite K-pop star answers my question quickly. | | | | | 0.721 | |
| The selection of celebrities is different compared to the contents of other platforms. | | | | | | 0.751 |
| It is unique compared to the video contents of other platforms. | | | | | | 0.841 |
| Naver V Live's video content differs from other platforms. | | | | | | 0.770 |
| Eigenvalue | 2.696 | 2.468 | 2.088 | 2.145 | 2.683 | 2.513 |
| % of Variance | 14.978 | 13.712 | 11.603 | 11.917 | 14.908 | 13.960 |
| Cumulative % of variance | 14.978 | 28.690 | 40.293 | 52.210 | 67.118 | 81.078 |

Prior to analyzing the mediating effect for RQ2 and RQ3, the results of the descriptive statistics of the variables are reported in Table 3. When using V Live, users have higher motivations in the following order: Social presence (M = 3.97, S.D. = 0.74), Differentiation (M = 3.41, S.D. = 1.00), Social interaction (M = 3.35, S.D. = 1.00), Social presence (M = 3.20, S.D. = 1.00), and Interpersonal relationship (M = 2.85, S.D. = 0.96). When viewing V Live, users' participation is ranked in the following order: Clicking "hearts" (M = 3.81, S.D. = 1.05), Commenting (M = 3.15, S.D. = 1.13), and Sharing video content (M = 2.68, S.D. = 1.18). The average weekly viewing time of V Live users was approximately 190 min (S.D. = 297.71), and the extent of fanship for a K-pop star was 3.28 out of 5 (S.D. = 0.87).

**Table 3.** Descriptive Statistics.

|  |  | **Mean** | **S.D.** |
|---|---|---|---|
| Independent variable * | Fanship | 3.28 | 0.87 |
| Mediator variable * | Celebrity presence | 3.97 | 0.74 |
|  | Celebrity support | 2.98 | 1.14 |
|  | Social presence | 3.20 | 1.00 |
|  | Interpersonal relationship | 2.85 | 0.96 |
|  | Differentiation | 3.41 | 0.83 |
|  | Social interaction | 3.35 | 1.00 |
| Dependent variable * | Weekly viewing time (min) | 190.13 | 297.71 |
|  | Commenting | 3.15 | 1.13 |
|  | Pressing hearts | 3.81 | 1.05 |
|  | Sharing | 2.68 | 1.18 |

* All variables were measured with Likert 5-point scale except for "Weekly viewing time".

The impact of the mediator on the independent and dependent variable is divided into direct and indirect effects. The direct effect is the effect of the independent variable on the dependent variable in the presence of the mediator (path c in Figure 1). The indirect effect is the effect of the independent variable on the dependent variable through a mediator (path a × b in Figure 1). The sum of direct and indirect effects is called the total effect. The total effect is the effect of an independent variable on a dependent variable when the mediator is not considered.

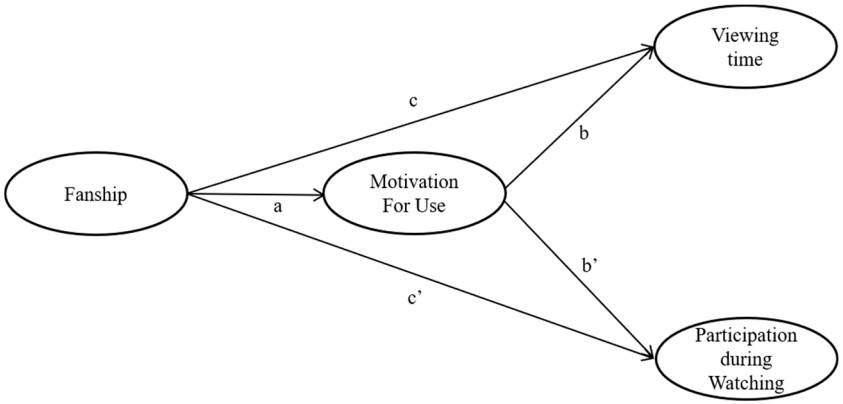

**Figure 1.** Research model.

In addition, a full mediating effect means a relationship in which an independent variable necessarily affects a dependent variable through a parameter; partial mediating effect indicates a relationship in which an independent variable can affect a dependent variable even if a mediator does not exist. The two conditions of partial mediating effect are as follows: (1) The impact of an independent variable on a mediator (path a in Figure 1), the impact of an independent variable on a dependent variable (path c in Figure 1), and the impact of a mediator on a dependent variable (path b in Figure 1) shall all be statistically significant; (2) The impact of an independent variable on a dependent variable (path c in Figure 1) shall be less than the total effect, which is the sum of the direct effect (c) and the indirect effect (a × b). To verify the direct and indirect effects and the partial mediating effect, we put six motivation factors as mediators into a single model and conducted three regressions according to Baron & Kenny (1986) and a Sobel test [26].

### 4.2. RQ2

Research question 2 is used to examine the relationship between fanship, motivation for using V Live, and the viewing time of V Live and to verify the mediating effect of motivation for using V Live in the relationship between fanship and viewing time.

Figure 2 shows the results for the mediating effect. The direct effect of fanship on viewing time was 0.264 ($p < 0.000$); the higher the fanship to K-pop stars, the more V Live users watch. Amongst six motivation factors, only the mediating effects of "Social presence" and "Differentiation" were statistically significant. With regards to "Social presence", the impact of fanship (IV) on social presence (Mediating effect; $\beta = 0.379$, $p < 0.000$), the impact of social presence on V Live viewing time (DV; $\beta = 0.148$, $p < 0.05$), and the impact of fanship on V Live viewing time ($\beta = 0.208$, $p < 0.01$) were all statistically significant. Moreover, the impact of fanship on V Live viewing time (direct effect) was less than the total effect (0.264); the total effect of 0.264 is the result of adding 0.208, the direct effect of fanship (IV) on V Live viewing time, and 0.056 ($0.379 \times 0.148$), the indirect effect of social presence (MV). Those results met the conditions of Baron & Kenny (1986) [26]. In case of "Differentiation", the impact of fanship (IV) on differentiation (Mediating effect; $\beta = 0.468$, $p < 0.000$), the impact of differentiation on V Live viewing time (DV; $\beta = 0.177$, $p < 0.05$), and the impact of fanship on V Live viewing time ($\beta = 0.181$, $p < 0.05$) were all statistically significant. Moreover, the impact of fanship on V Live viewing time (direct effect) was less than the total effect (0.264); the total effect of 0.264 is the result of adding 0.181, the direct effect of fanship (IV) on V Live viewing time, and 0.083 ($0.468 \times 0.177$), the indirect effect of differentiation (MV). All the conditions of Baron & Kenny (1986) were satisfied [26]. In other words, the "Social presence" and "Differentiation" motivation factors of V Live mediated the relationship between fanship and V Live viewing time, and a partial mediation effect occurred on the relationship between each motivation factor and viewing time.

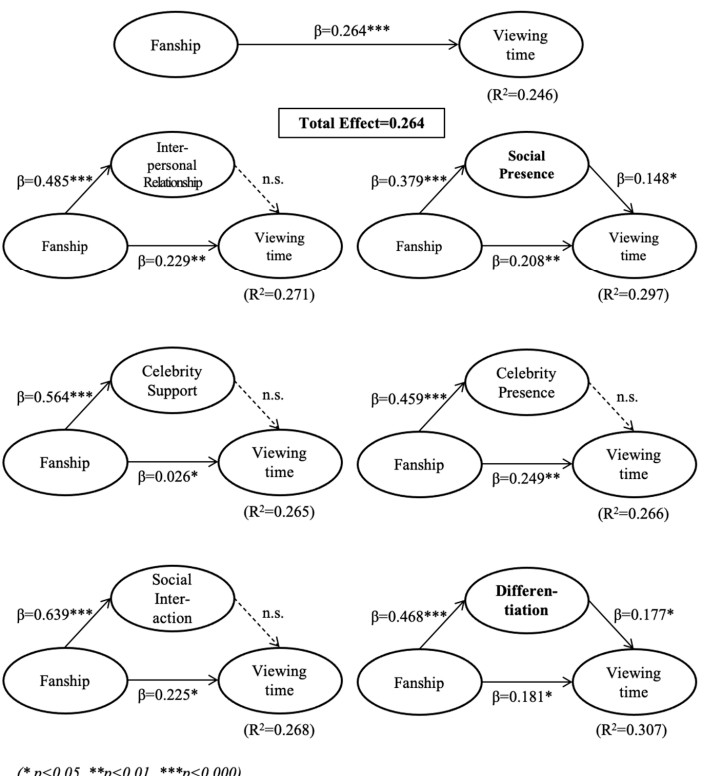

**Figure 2.** Mediating Effect of Use motivation between Fanship and Viewing time.

As a result of the Sobel test, the value of "Social presence" was 2.93 ($p < 0.05$), and that of "Differentiation" was 3.61 ($p < 0.000$); as the value of Sobel test is higher than 1.96, the mediating effect

of each motivation for using V Live has been verified. The higher the fanship for K-pop, the more likely are fans to feel as if they are in the same location as other fans (Social presence), and the higher the motivation to use the service exclusively provided on V Live (Differentiation), the more V Live users who watch the video content.

### 4.3. RQ3

The research question 3 is to identify the relationship between fanship, motivation for using V Live, and the extent of participation while watching V Live; to verify the mediating effect of motivation for using V Live in the relationship between fanship and the extent of participation during watching V Live.

The impact of fanship on the extent of participation during watching was 0.600 ($p < 0.000$); The higher the fanship for K-pop stars, the more hearts clicked, the more comments posted, and the more video content shared.

Figure 3 and Table 4 indicate that the mediating effects of all each motivation factors were statistically significant; the impact of fanship (IV) on the motivation factors (MV), the impact of fanship on the extent of participation during viewing V Live (DV), the impact of motivation factors on the extent of participation during viewing were all significant. Moreover, all the direct effects of six motivation factors were less than the total effect. In other words, the relationship between fanship and participation was mediated by motivation for use: "Interpersonal relationship", "Social presence", "Celebrity support", "Celebrity presence", "Social interaction", and "Differentiation". All the conditions for Baron & Kenny (1986) were satisfied [26]. A partial mediating effect was found on the relationship between each motivation factor and participation.

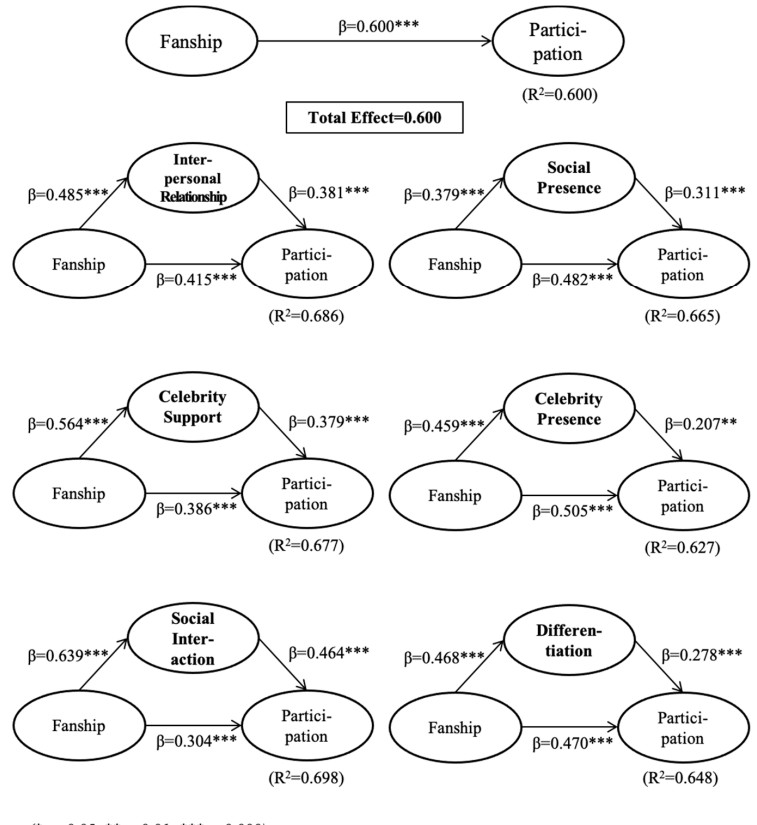

**Figure 3.** Mediating Effect of Use motivation between Fanship and Participation during Watching.

**Table 4.** Descriptive Statistics.

| | Direct Effect (Path c) | Indirect Effect (Path a × b) | Total Effect (t) |
|---|---|---|---|
| Fanship→Interpersonal relationship→Participation | 0.415 | 0.185 (0.485 × 0.381) | |
| Fanship→Social presence→Participation | 0.482 | 0.118 (0.379 × 0.311) | |
| Fanship→Celebrity support→Participation | 0.386 | 0.214 (0.564 × 0.379) | 0.600 |
| Fanship→Celebrity presence→Participation | 0.505 | 0.095 (0.459 × 0.207) | |
| Fanship→Social interaction→Participation | 0.304 | 0.296 (0.639 × 0.464) | |
| Fanship→Differentiation→Participation | 0.470 | 0.130 (0.468 × 0.278) | |

* Total effect (t) = Direct effect (c) + Indirect effect (a × b).

As a result of the Sobel test, the value of "Interpersonal relationship" was 6.35 ($p < 0.000$), "Social interaction" was 8.72 ($p < 0.000$), "Celebrity support" was 7.24 ($p < 0.000$), "Celebrity presence" was 5.12 ($p < 0.000$), "Social interaction" was 8.72 ($p < 0.000$), and "Differentiation" was 5.65 ($p < 0.000$); as all the values of the Sobel test were higher than 1.96, the mediating effect of each motivation for using V Live has been verified.

In other words, the more passionately V Live users love their own favorite K-pop stars (i.e., the higher the fanship), the higher the motivation for using V Live. Further, the higher the motivation for using V Live, the more "hearts" were clicked, comments were posted, or video content was shared.

## 5. Conclusions and Discussion

This study investigated the motivation for using V Live, a previously unexplored aspect. Six factors were derived as motivation for using V Live. V Live is used to build a relationship with people or other fans (Interpersonal relationship); to feel as if they are in the same place with other users (Social presence); to boost the popularity of their favorite K-pop stars (Celebrity support); to feel as if they are in the same place with their favorite K-pop star (Celebrity presence); to interact with favorite K-pop stars (Social interaction); to use a differentiated service (Differentiation).

Overall, the findings indicate a positive relationship between fanship and viewing time, or fanship and the extent of participation during watching; the more enthusiastically the V Live users love their favorite K-pop stars (the higher the fanship), the more they watched V Live and participated in V Live content while viewing it by clicking "hearts", commenting, or sharing video content. It was observed that the more V Live users a K-pop star had who were big fans, the more these users tended to watch video content starring the K-pop star; and this increased their likelihood of clicking "hearts", commenting, or sharing video content. Moreover, the mediating effect of motivation for using V Live on each relationship was positively significant. First, with regards to "Social presence", the more V Live users a K-pop star had that were big fans, the more these users tended to feel the existence of other fans even in the virtual space; the higher the motivation for Social presence was, the more the users watched video content, clicked "hearts", commented, or shared video content. This is consistent with the results of Reysen et al. (2017): the stronger the users' identity as a K-pop star's fan, the higher the motivation to build social relations with other fans, and the greater the users' satisfaction of their experience with other users increased [27]. In other words, the more enthusiastic users are for a K-pop star, the more they perceive V Live (a virtual space) as a physical space with other fans, and ultimately experience more video content on V Live.

Second, in terms of "Differentiation", the more the V Live users loved a K-pop star, the more these users tended to use V Live as an absolutely differentiated service from other services; the more the motivation of Differentiation was, the more content was watched, "hearts" were clicked, comments were posted, or video content was shared by users. V Live offers various video content ranging from free-style live streaming content to professionally produced high quality video, as well as original content with a novel theme [13]. For example, "Run! BTS", a variety show by BTS, the popular boy band, is original content exclusively provided on V Live; a total of 85 episodes have been uploaded since 1 August 2015 (the release date of V LIVE). The theme of the show varies every other week, for example, BTS fans can see the band members offstage, traveling, or playing games. Therefore,



V Live's original content that is exclusively offered accurately targeted fans' loyalty to idols [28]. In other words, the more the users love a K-pop star, the more these users tend to see various aspects of their favorite K-pop stars; the higher the motivation of Differentiation, the more time spent watching video content on V Live; the more "hearts" they clicked, comments they wrote, or video content they shared to express their satisfaction.

Third, in the case of "Celebrity presence", the more the V Live users like their K-pop stars, the more motivated they are to feel like they are in the same place as their friends (Celebrity presence); the higher the motivation of Celebrity presence, the more the participation during watching V Live. In other words, the more enthusiastic the fans were, the more motivated they were to search for the natural look of their favorite K-pop stars while feeling like they were close together; they express positive emotions more passionately through the number of "hearts" or comments. Lee & Jang (2011) found that Twitter users felt more strongly as if they had a direct conversation with a celebrity after scrolling through the celebrity's Twitter account, leading to users being more interested in the celebrity and more willing to watch a TV program or a movie that the celebrity appears in [29]. A sense of closeness was formed simply by looking at the celebrity's Twitter account. Hu & Chaudhry (2020) verified that social bonds had positive impact on the affective commitment to the streamer, and the affective commitment induced the user's engagement in e-commerce live streaming [22].

As prior studies, it can be expected that the bond formed by communicating with K-pop stars directly in real-time through V Live will be much stronger. Most V Live users are avid fans of K-pop stars, which will increase their motivation to feel as if K-pop stars are close friends or to build bonds through directly communicating on V Live with their favorite K-pop stars; the motivation to build close bonds with K-pop stars will lead to more active participation while watching V Live. Also, if a star desires to build emotional ties with his/her fans, live streaming can be used as a useful tool. Especially in freestyle live streaming (Ham & Lee, 2020) [13], if a star's appearance is shown in a natural way, fans will feel more familiar with him/her.

Fourth, in the case of "Interpersonal relationships", the more enthusiastically V Live users loved K-pop stars, the higher the incentive to build relationships with people around them or other fans (Interpersonal relationship); the higher the motivation, the more actively they participated in watching V Live. Fans usually consume video content by using multiple platforms at the same time, not only one platform. In other words, fans use V Live to watch live-streaming content of their favorite K-pop stars, but not only V Live for their fan activities. The online fan community, Twitter, and Instagram are simultaneously used to obtain information related to favorite K-pop stars or communicate with other fans to form ties [30]. Prior to and during K-pop star live-streams, fans encourage other fans to press "hearts" on V Live on online fan communities or on Twitter to increase the number of hearts. After the live-streaming ends, fans re-edit the full-version live-streaming content into video clips, share, and replay the highlights on SNS with other fans. With the advent of V Live, fans are able to expand the scope of online fan activities. In particular, unlike online fan communities or Twitter which are used in a natural way, V Live provides original content for a majority of K-pop fans. V Live provides an online virtual space where countless fans from around the world can gather in one space and communicate about what they like. On V Live, fans produce additional values by utilizing their own language skills such as spontaneously translating for other fans.

Fifth, in terms of "Celebrity support", the higher the fanship of V Live users for K-pop stars, the more motivated they are to boost the popularity of their favorite K-pop stars; the higher the motivation, the more they comment, press hearts and share video contents. Unlike other LSs, V Live holds its own awards ceremony, "Global Artist Top 10", and gives prizes to the top 10 teams by calculating the number of "hearts", comments, plays, and the attendance score of fans on all video contents posted for a year on each star's channel. Such an award ceremony is an opportunity to officially prove the popularity of K-pop stars and the teamwork of fans. The higher the fanship of V Live users, the stronger their desire to prove the K-pop star's popularity, which can naturally lead to watching and engaging in V Live content. It is also noteworthy that the unique feature that the "heart"

of V Live can be pressed indefinitely formed a unique fandom culture called "Heart labor". On the other hand, the result suggests that if a manager of LS is considering how to make a user stay longer in service, he/she can come up with various ways to enhance user's participation such as "Global Artist Top 10".

Sixth, when the dependent variable was viewing-time, the impact of motivation for using V Live on viewing-time was not statistically significant, except for "Social presence" and "Differentiation". This means that simply watching video content for a long time in an online space where K-pop stars and users directly communicate is not a big driver for V Live users. The benefit of using V Live is "Global Artist Top 10", which is selected by V Live once a year, and the selection is based on the combined score of "hearts", "comments", "plays" and "fan attendance". In addition, as LS users prove their existence by sponsoring virtual money to streamers [8], V Live users engage in active participation rather than watching as a means of expressing their love for K-pop stars and their video contents.

This study has a few limitations. First, as the measurement for participation, we adopted the extent of commenting, pressing "hearts", and sharing, which can be done for free without having to consider purchasing paid content. However, users who pay for V Live content may be the most avid fans of K-pop stars. Considering this, the frequency of purchase and the amount of paid content for V Live will be measured in a follow-up study. Second, according to Naver, 80% of all V Live users are overseas [31]. However, the survey for this study was conducted on domestic users in South Korea, and it is difficult to generalize the findings for overseas users. Based on this study, the usage behavior of overseas V Live users needs to be considered in the future.

The implications of this study are as follows. First, this study is meaningful because it revealed motivation factors that encourage V Live users to watch and participate, which were previously unexplored. We identified the motivation for "Celebrity support", a fundamental driver for the new fandom culture—"Heart labor", and "Celebrity presence", in which V LIVE users felt the presence of K-pop stars through live-streaming content.

Second, this study suggested the potential of LS to function as a social community. V Live users felt that they were physically in the same space as other users, and enjoyed seeing the number of "hearts" constantly rising as well as comments uploaded simultaneously with the video content played. This indicates that V Live not only enables real-time interactions between K-pop stars and fans but also brings together fans from around the world in an online space and strengthens fandom solidarity.

Third, differentiated services such as exclusive video content can increase satisfaction for the service and become a stable revenue source for LS platforms. The more the fanship for streamers, the higher the motivation to use differentiated services that fans could not previously experience; the higher the motivation, the more video content users watch and participate in while watching. Although the purchase of paid content was not considered in this study, video content consumption is also interpreted as a kind of participation according to prior studies. Fans with high fanship for streamers will be loyal to streamers and will be willing to pay for paid content. The mechanism from fanship on the motivation of differentiation to viewing time/participation was identified, and it suggests that the more differentiated the service provided by V Live, the higher the revenue generated, and that the strategy to provide differentiated services such as exclusive video content—especially for users with high fanship—can be effective.

**Author Contributions:** Conceptualization, methodology, writing—review & editing: S.W.L., Data collection, data analysis, writing—original draft: M.H. All authors have read and agreed to the published version of the manuscript.

**Funding:** This research received no external funding.

**Acknowledgments:** This research was supported by the MSIT (Ministry of Science and ICT), Korea, under the ITRC (Information Technology Research Center) support program (IITP-2020-0-01749-001) supervised by the IITP (Institute of Information & Communications Technology Planning & Evaluation).

**Conflicts of Interest:** The authors declare no conflict of interest.

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
