# Peer review of "Factors Influencing Viewing Behavior in Live Streaming: An Interview-Based Survey of Music Fans"

_mti, doi:10.3390/mti4030050_

Round 1

Reviewer 1 Report

The authors have addressed a timely research topic. The paper is well written. However, the literature review is rather limited and focuses on a limited International journals. The authors need to expand this section of the paper. For example, a recent paper M. Hu and S.S. Chaudhry, Enhancing consumer engagement in e-commerce live streaming via relational bonds, Internet Research, https://doi.org/10.1108/INTR-03-2019-0082. Also, the authors need to provide additional insights and specific strategies/implications for the broadcasters of live-streaming services based on the outcomes of their research.

Author Response

Point 1: The authors have addressed a timely research topic. The paper is well written. However, the literature review is rather limited and focuses on a limited International journals. The authors need to expand this section of the paper. For example, a recent paper M. Hu and S.S. Chaudhry, Enhancing consumer engagement in e-commerce live streaming via relational bonds, Internet Research, https://doi.org/10.1108/INTR-03-2019-0082.

Response 1: Thanks for your opinion. We added three papers on the section of “2. Literature review,” including that you recommended.

  • Hu, M., & Chaudhry, S. S. (2020). Enhancing consumer engagement in e-commerce live streaming via relational bonds. Internet Research, 30(3),1019-1041.
  • Wan, J., Lu, Y., Wang, B., & Zhao, L. (2017). How attachment influences users’ willingness to donate to content creators in social media: A social-technical systems perspective. Information & Management, 54, 837-850.
  • Hou, F., Guan, Z., Li, B., & Chong, A. Y. L. (2020). Factors influencing people’s continuous watching intention and consumption intention in live-streaming. Internet Research, 30(1), 141-163.

Also, for more generalization, we revised the title of this paper as “Factors influencing viewing behavior in live streaming: An interview-based survey of Music fans”

Point 2: Also, the authors need to provide additional insights and specific strategies/implications for the broadcasters of live-streaming services based on the outcomes of their research.

Response 2: As you suggested, we stated additional strategies for the broadcasters and the managers of LS in the section “5. Conclusions and Discussions.”

Reviewer 2 Report

Overall, this was well-written and an apt study given the small amount of research with this fan group. I only have one concern: instead of conducting separate mediations for each motivation, why not throw all the motivations into a single mediation model? I suspect you will find that some motivations are better than others (i.e., some may not be significant). In other words, if you put them all into one model you can let the motivations compete to see which ones are stronger when accounting for the others. You can run such a mediation in Andrew Hayes’ PROCESS macro (https://www.processmacro.org/index.html). Beyond this, the paper is well done.

Author Response

Point 1: Overall, this was well-written and an apt study given the small amount of research with this fan group. I only have one concern: instead of conducting separate mediations for each motivation, why not throw all the motivations into a single mediation model? I suspect you will find that some motivations are better than others (i.e., some may not be significant). In other words, if you put them all into one model you can let the motivations compete to see which ones are stronger when accounting for the others. You can run such a mediation in Andrew Hayes’ PROCESS macro (https://www.processmacro.org/index.html). Beyond this, the paper is well done.

Response 1: Thanks for your opinion. The analyzed model of this study was a full mediation model with six mediators (six motivation factors such as Interpersonal relationship, social presence, celebrity support, celebrity presence, social interaction, differentiation). In other words, we put all the motivations into a single mediation model, conducted the three-step regression analysis according to Baron & Kenny (1986). All models were presented separately to reduce complexity. But we understand that the models of Figure 2 and Figure 3 could be confused as each separated model. So, we added a description of how the research model was analyzed above the section “4.2. RQ2.”
